# Comparison of Robot-Assisted and Manual Cannula Insertion in Simulated Big-Bubble Deep Anterior Lamellar Keratoplasty

**DOI:** 10.3390/mi14061261

**Published:** 2023-06-16

**Authors:** Yinzheng Zhao, Anne-Marie Jablonka, Niklas A. Maierhofer, Hessam Roodaki, Abouzar Eslami, Mathias Maier, Mohammad Ali Nasseri, Daniel Zapp

**Affiliations:** 1Klinik und Poliklinik für Augenheilkunde, Technische Universität München, 81675 München, Germany; yinzheng.pro@gmail.com (Y.Z.); ajablonka@hotmail.de (A.-M.J.); niklas.maierhofer@mri.tum.de (N.A.M.); mathias.maier@mri.tum.de (M.M.); daniel.zapp@mri.tum.de (D.Z.); 2Translational Research Lab, Carl Zeiss Meditec AG, 81379 München, Germany; he.roodaki@tum.de (H.R.); abouzar.eslami@zeiss.com (A.E.)

**Keywords:** robotic assistance, intraoperative OCT, DALK

## Abstract

This study aimed to compare the efficacy of robot-assisted and manual cannula insertion in simulated big-bubble deep anterior lamellar keratoplasty (DALK). Novice surgeons with no prior experience in performing DALK were trained to perform the procedure using manual or robot-assisted techniques. The results showed that both methods could generate an airtight tunnel in the porcine cornea, and result in successful generation of a deep stromal demarcation plane representing sufficient depth reached for big-bubble generation in most cases. However, the combination of intraoperative OCT and robotic assistance received a significant increase in the depth of achieved detachment in non-perforated cases, comprising a mean of 89% as opposed to 85% of the cornea in manual trials. This research suggests that robot-assisted DALK may offer certain advantages over manual techniques, particularly when used in conjunction with intraoperative OCT.

## 1. Introduction

Corneal pathologies such as advanced keratoconus, stromal dystrophies, or central scarring of the deeper layers of the corneal stroma present typical indications for corneal transplantation. Penetrating keratoplasty (pKP) replaces all corneal layers down to and including the endothelium and oftentimes remains the only therapeutic option for deep central traumatic scars, dystrophies with full-thickness corneal involvement, or ulcerations up to perforation [1,2]. However, if the disease is confined to the epithelium and/or stromal corneal layers, lamellar keratoplasty can be applied. Yet lamellar anterior grafts have been shown to be inferior in terms of best achievable visual acuity if the transplanted stromal tissue interfaces with remnants of the recipient’s own stromal layers. Out of these findings, surgical techniques have been refined to successfully perform the so-called Deep Anterior Lamellar Keratoplasty (DALK). DALK is a surgical technique that preserves the host Descemet’s Membrane (DM) and endothelial layer while trying and separating both from any remnants of stromal layers on top, given these deepest layers are healthy and intact. The big-bubble technique, as first described by Anwar et al. in 2002 [3], is the most commonly used and likely most standardized technique, using different air or liquid injections into the intersection of stromal and descement layers to achieve mechanical separation before excision of the former.

By preserving the host’s own endothelium and ideally keeping the anterior chamber uncompromised, DALK has been shown to be associated with significantly less endothelial graft rejection compared to pKP [4,5,6]. Additionally, secondary glaucoma has been observed less frequently in DALK as compared to classical pKP [7], while acuity-wise, unrejected grafts [5] following DALK have exhibited comparable graft clarity and visual acuity outcome [8,9] in comparison to pKP. However, the crucial step in successfully performing prescribed DALK is precise air injection at a sufficient depth, with research suggesting successful bubble formation at a depth traversing ≥80% of the corneal stromal thickness [10,11,12], while depths greater than 90% are believed to be beneficial. Naturally, these are associated with greater risk of DM perforation and make DALK a technically demanding procedure with a relatively high failure rate. Intraoperative conversion to pKP is necessary in such cases [5,7,13]. Conversion to pKP reportedly ranges between 4% and 39.2% in big-bubble DALK procedures [14,15,16]. (One more recent study reported a relatively high conversion rate of 42.5% [17]).

Two key parameters can be changed in order to facilitate cannula insertion to sufficient depth. Firstly, depth perception throughout the procedure can be enabled via Optical Coherence Tomography (OCT); secondly, accuracy can be improved by utilizing robotic assistance to manage tool tip oscillation, which can occur due to physiological hand tremor.

In several studies, intraoperative OCT has been used for cannula monitoring throughout the big-bubble DALK procedure, allowing for higher depth perception accuracy [18,19]. Draelos et al. examined the combination of intraoperative OCT and robotic assistance during big-bubble DALK cannula insertion in two recent studies [20,21]. Insertions were performed by inexperienced surgeons on ex vivo human corneas. It was found that the use of a cooperative system allowed for a 31% increase in reached depth compared to manual trials. In addition, perforation rates in the cooperative trials were lower compared to manual trials without image guidance.

In this work, the feasibility and implications of using a surgical robot for the cannula insertion phase of DALK were studied. To achieve this, a custom-built robot in a master–slave setup was utilized. Simulated DALK procedures were performed on ex vivo porcine eyes using surgical microscopes equipped with intraoperative continuous OCT capture capabilities. As opposed to the setup used by Draelos et al. [20], intraoperative OCT was implemented throughout both manual and robotic trials, enabling the surgeons to intraoperatively assess cannula depth. Furthermore, each cannula insertion was followed by air injection to immediately evaluate the airtightness of the formed tunnel and demarcation of the achieved injection plane by OCT. While the approach by Draelos focused on the reached depth without perforation by performing up to 8 insertions into each cornea, this work aimed to evaluate successful depth achievement of the demarcated plane at over 80% of the stromal thickness. Quantitative measures such as the number of perforations and the reached stromal depth were assessed for comparison.

## 2. Materials and Methods

### 2.1. Microscope and Intraoperative OCT

A standard ophthalmic surgery microscope equipped with an OCT engine (LUMERA 700 with RESCAN 700, Carl Zeiss Meditec, München, Germany) was used for all experiments. Two OCT B-scans in a cross formation were continuously captured by the microscope at a desired location controlled by the surgeon using a foot control pedal or by the assistant using the auxiliary screen. The surgical field was directly observed through the microscope’s eyepiece with the two B-scans and an OCT acquisition location marker overlaid onto the scene.

### 2.2. Robot

A custom-made hybrid parallel–serial robot with prismatic piezo actuators was used for the robotic trials (Figure 1). The robot had five degrees of freedom and was controlled in a master–slave fashion using a 3D mouse in these experiments. The syringe was mounted on the last joint of the robot. Operation of the robot was internally handled through optimized collective movements of joints. The working volume of the robot was almost a perfect sphere with a radius of 30 mm; this working volume could support the entire corneal area, if the robot was placed appropriately. The robot was controlled by the joystick based on the velocity control, which showed that the more value created from the joystick, the higher the speed of the robot. This mechanism guaranteed a safe relative motion of the tool. Moreover, due to the fact that we used velocity control, direct motion scale was not necessary while the image was observed from the microscope.

### 2.3. Instruments

A rubber base imitating the orbit was used to secure porcine eyes throughout the procedure. Pins were used to fixate eyes by the lateral conjunctive tissue onto the base to enable passive movements simulating elevation and depression along the transversal axis. Porcine eyes were obtained from a local slaughterhouse and used for the experiments within 3 h after the collection. For standardization of intraocular pressure, a 23-gauge trocar was implanted in the vitreous cavity and homogenous pressurization of the globe to around 20 mmHg was ensured by passive infusion of balanced salt solution by bottle height. For insertion and injection, 1 mL syringes were tipped with sharp disposable 30 G cannulas of 12 mm length.

### 2.4. Surgical Technique

Big-bubble pneumodissection technique in DALK aimed for posterior stroma detachment by air injection at a sufficient depth of at least 80%, causing an air bubble to form between DM and posterior stroma [11]. If the air bubble formation succeeded, access to and removal of stroma located anterior to DM was facilitated. Typically, the procedure started with the insertion of a thin 27 G or 30 G straight or bent cannula into the host cornea with its beveled side facing towards the DM. Traversing the cornea, the cannula produced an airtight tunnel. At the desired depth, air was injected into the deep corneal stroma leading to detachment of posterior stroma from DM and endothelium. While preserving the DM and endothelium, the remainder of the corneal tissue was removed and replaced with a donor graft [3,22].

### 2.5. Method

Two novice surgeons with no prior experience in DALK were trained to perform the procedure by receiving verbal instructions and supervision from an experienced ophthalmic surgeon. Thereafter, each surgeon performed manual and robot-assisted DALK practice trials on ex vivo porcine corneas. The main big-bubble DALK cannula insertion followed by air injection experiments was simulated on ex vivo porcine eyes. Although the actual detachment of the stroma from DM in these specimens was not reproducible due to the typically young age of the pigs with stronger adhesion strength, thinner thickness of DM and immature tissue, a porcine model was chosen for two reasons. Firstly, parameters ensuring high chances of successful bubble formation and pneumodissection in DALK were well-documented and could be replicated reliably in porcine models. Secondly, a large sample size could be gathered without potential waste of human donor material in our proof-of-concept setup with a custom-made robot in a newly established OCT-robot assistance setup.

Operators were randomly assigned to trial order and setting and the other operator assumed the assistant role. This showed that the order in which the surgeons performed the manual and robot-assisted trials was randomized, which might minimize any potential bias due to learning effects or other factors. For each manual and robot-assisted trial, a 1 mL syringe filled with air was tipped with a disposable sharp 12 mm long 30 G cannula. In half of the trials, the air-filled syringe was held and guided manually by the surgeon. In this setting, the cannula was bent approximately at the base at an angle of about 60 degrees for better ergonomics corresponding to the state of the art [22]. In the other half of the trials, the air-filled tipped syringe was mounted onto the robot, which was then remotely controlled by the surgeon. In this setting, bending of the 30 G cannula was omitted since no effect on ergonomics could be achieved.

The point of entrance within the cornea was chosen close to the limbus to ensure sufficient tunnel length associated with a higher chance of airtightness. In robot-assisted trials, the robot with the syringe mounted onto it was positioned to a distance of around 2 cm close to the intended entrance point. The initial positioning of the robot in the experimental setup was performed by setting the joints in the reference point by a calibration algorithm and bringing the robot manually to the insertion point before the surgeon kept controlling the robot by the joystick when the needle was in contact with the tissue. However, in our subsequent work, we utilized an augmented reality method to place the robot in a position to optimize the working volume, making sure all target areas were reachable. Intraoperative OCT was enabled and focused by the assistant so both the corneal stroma and the cannula were displayed in cross- and longitudinal sections. The cannula was then aligned following the intended insertion angle relative to the corneal central surface. The alignment was realized by remote-controlled fine adjustments continuously carried out by the assistant. Insertion was executed aiming for an estimated depth of 90% or greater while taking care to avoid penetration into the DM (Figure 2). Depth information was continuously obtained from projected real-time intraoperative OCT images and, on inquiry, confirmed or reevaluated by the assistant. At the desired depth, air injection was performed, and formation of the demarcation plane was observed and recorded.

### 2.6. Parameters

Real-time intraoperative OCT and microscopic visual field were recorded throughout all trials. Three-dimensional OCT volumes of the cannula tip and the corneal stroma were captured at multiple milestones: prior to cannula insertion, prior to air injection at the desired depth, post-air injection displaying demarcation plane representative of potential successful bubble formation, and post-cannula retrieval displaying the caused tunnel and the final outcome. Success or failure was determined based on the outcome of the injection. Failed trials consist of unsuccessful demarcation plane due to air leakage, formation of superficial small bubbles rather than a demarcation line, and perforations in which the cannula tip punctures the DM at any point in time along the procedure. Failed big-bubble formation was indicated by the observation of small air bubbles throughout the entire stromal layer [23,24] instead of a homogenous demarcation layer. Superficial intrastromal air was reflecting the OCT rays and thus appeared as a cloudy layer directly below the corneal epithelium, allowing for clear differentiation from the desired deep stromal to predescemental demarcation (Figure 3). In supposedly successful bubble formation, however, the injected air caused a sharp homogenous demarcation layer right above the DM without penetrating into more superficially located stromal layers (Figure 4). Additionally, the insertion angle, tunnel length, and reached depth were measured. The insertion angle referred to the angle comprising the longitudinal section of the cannula and corneal central surface prior to insertion. It was calculated by analyzing the volumetric OCT captures immediately before insertion. The tunnel length referred to the length of the tunnel produced by the inserted cannula from point of entry at the corneal surface along the cannula down to the tip of the bevel and was calculated by analysis of the volumetric OCT captures immediately before injection. The reached depth was calculated via analysis of volumetric OCT captures immediately before insertion and immediately after cannula retraction in successful trials. It corresponded to the deepest position the needle tip had reached in the cornea during the procedure and was reported in percentage of corneal stroma traversed by the cannula tip (Table 1).

## 3. Results

DALK cannula insertion was simulated on 48 ex vivo porcine eyes obtained from healthy rearing pigs. The corneal thickness of each eye was measured using OCT. The mean observed corneal thickness was 998 μm with a standard deviation of 105 μm. Due to low intraocular pressure in two eyes despite passive intravitreal infusion, only 46 trials, 23 manual and 23 robot-assisted, were included in the evaluation. Success and failure rates are shown in Table 1. Although in robotic trials, less perforation occurred (one versus three cases in manual trials), failure due to unsuccessful generation of a sharp, sufficiently deep demarcation layer was present in both settings. Both the differences in perforation rates as well as in success rates in robotic and manual trials were not significant (two-tail *t*-test with significance of *p* < 0.01).

The mean tunnel length in all successful trials was 1.76 mm with a standard deviation of 0.38 mm and only differed slightly in robotic (mean 1.72 mm, standard deviation 0.45 mm) and manual (mean 1.82 mm, standard deviation 0.27 mm) trials. The mean reached depth in successful cases was greater in robotic trials (89.29% of the cornea with a standard deviation of 4.01%) than in manual trials (85.05% of the cornea with a standard deviation of 4.52%). This difference was significant (independent samples *t*-test with significance of *p* < 0.01). The reached depths are plotted in Figure 5. In all cases in which no perforation occurred, an average volume of 0.24 mL of air was injected. The mean duration of surgery in successful cases performed with robot assistance was 261 s with a standard deviation of 96 s, which was considerably longer than manual surgery, at 212 s with a standard deviation of 96 s. Surgical time did not include robot preparations and calibrations.

## 4. Discussion

This work showed that novice surgeons with no prior experience in performing DALK were capable of generating an airtight tunnel in the porcine cornea under the given conditions, resulting in the successful generation of a sharp, deep stromal demarcation plane representing sufficient depth reached for big-bubble generation in most cases. The combination of intraoperative OCT and robotic assistance in this study resulted in a significant increase in the depth of achieved detachment in non-perforated cases, consisting of a mean of 89% as opposed to 85% of the cornea in manual trials. As a reached depth of greater than 79% has been reported to likely result in successful bubble formation, the mean depths reached in this study both in manual and robotic trials were to be considered sufficient to assume a high probability of full pneumodissection in human eyes. Pasricha et al. [11] reported that depths surpassing 90% were not conditional for successful pneumodissection. However, depths up to a maximum of 96% were reached in robotic trials without perforation. It was conceivable that incorporating robotic assistance in DALK or similar ophthalmic procedures may be beneficial by increasing the accuracy without compromising safety.

The mean reached depth in the robot-assisted setting in the performed experiments was greater (89%) than the average depth reported in an equivalent study (71%) conducted by Draelos et al. As in their study, where air injection did not follow the cannula insertion, it could not be assessed whether perforation would have occurred during or after the injection. It was important to note that the mean depth measured in this study only consists of cases in which the air injection was successfully evaluated by the analysis of OCT scans captured after air injection. By doing so, the tunnel could be proven to be airtight, indicating a precise advancement of the cannula. However, as can be observed in Figure 4, DM detachment was not visible despite successful bubble formation, which concurred with previous observation in porcine models, with porcine eyes typically having very strong debasement layer adhesions due to the young age at which rearing pigs were slaughtered. Tunnel lengths were assessed after injection showing a mean length of 1.76 mm in successful trials, not differing significantly from tunnel lengths in failed cases with a mean of 1.88 mm. However, the sample size of failed cases was too small to study potential differences in tunnel lengths. To our knowledge, the tunnel length in DALK has not yet been investigated beyond the natural necessity of being sufficiently airtight for pressure build-up, making it a topic of interest for further research.

The study focused on comparing the efficacy of robot-assisted and manual cannula insertion in simulated big-bubble deep anterior lamellar keratoplasty (DALK) using novice surgeons with no prior experience. The results showed that in successful cases performed with robot assistance, the mean duration of surgery was considerably longer at 261 s, with a standard deviation of 96 s. In comparison, manual surgery had a mean duration of 212 s, also with a standard deviation of 96 s. Surgical time did not include robot preparations and calibrations. It is important to note that the standard deviation is a measure of the amount of variation or dispersion in a set of data. The standard deviation of the operation time of the two methods was the same, and the average operation time of the robot-assisted system was longer than that of the manual operation. The main reason was that novice surgeons may require more time to become proficient in using robotic systems compared to traditional manual techniques. It was possible that differences in skill level could contribute to differences in operation time between manual and robot-assisted trials. The success rate of the procedure is a more major concern than the length of the procedure. Firstly, the operative time to perform DALK manually has increased compared to conventional penetrating keratoplasty. Secondly, as shown above, even in experienced hands, the associated conversion rate due to intraoperative penetration of the descement has historically prevented corneal surgeons from going through the additional efforts of DALK more often. Additionally, the remaining interstitial thickness thinning usually performed prior to the actual needle insertion does not require dissection of the uppermost layer, which may partially compensate for the longer time required for the actual needle insertion. Therefore, a system that offers higher success rates may improve acceptance rates, even if it involves longer preparation times to some extent. However, the combination of intraoperative OCT and robotic assistance received a significant increase in the depth of achieved detachment in non-perforated cases, consisting of a mean of 89% as opposed to 85% of the cornea in manual trials. While the study did find that robotic assistance combined with intraoperative OCT resulted in a significant increase in the depth of achieved detachment compared to manual trials, further research is needed to investigate factors that could impact operation time and their potential impact on surgical outcomes.

Although the combination of intraoperative OCT and robotic assistance in this study has shown a significant increase in the depth of achieved detachment in non-perforated cases, there are also potential complications and risks with its adoption. Firstly, there will be a longer study period for surgeons who have not used the robotic assistance system or are not familiar with it, which could increase risks of complications during the initial phase of using the system. Additionally, robot-assisted experiments also have the potential to fail. Failed trials include superficial small bubbles instead of a big-bubble formation, incomplete demarcation layer due to insufficient depth of needle, or air leakage. Finally, the cost of equipment maintenance for the robot-assisted system is also a problem that cannot be ignored, and it may not be affordable for some medical institutions.

It is essential to consider the following aspects regarding the validity of the study. Firstly, ex vivo porcine eyes instead of human eyes were used. It is known that the constitution and properties of the porcine cornea do not fully resemble those of the human cornea. Moreover, it is impossible to generate a true bubble formation in these specimens for the reasons given above. Therefore, repeating this successful ex vivo proof-of-concept with human corneas [25], similar to the setting of the work by Draelos et al., is favorable, now that a successful combination of our custom-built robot with high resolution iOCT depth information has shown to be advantageous. Secondly, as indicated by previous studies, the usage of intraoperative OCT provided a fairly accurate estimation of the cannula tip location during any point in the procedure. Inexperienced surgeons have been demonstrated to successfully perform simulated DALK guided by intraoperative OCT [26]. Successful cannula insertion should have likely resulted from obvious precise visualization provided by the intraoperative OCT feedback [20]. However, improved tremor management and ergonomics from employing surgical robots hold the potential to further improve success rates in such delicate surgeries as DALK of the cornea.

## 5. Conclusions

In summary, the application of robotic assistance allowed for significantly greater depth achievement in DALK without compromising safety. The study of this work suggested that in order to reliably reach a sufficient depth of at least 80% of the cornea, intraoperative OCT along with robotic assistance was feasible and beneficial, especially for novice or inexperienced surgeons. Robot-assisted surgical procedures can provide precise and accurate movements while maintaining stability during the surgical process. When combined with intraoperative Optical Coherence Tomography (OCT), robot-assisted systems enable visualization of surgical parameters, greatly enhancing the success rate of the surgery. Further potential applications of the designed robot and its proven increased precision such as assisting in subretinal injections are interesting and evolving fields for investigation.

## Figures and Tables

**Figure 1 micromachines-14-01261-f001:**
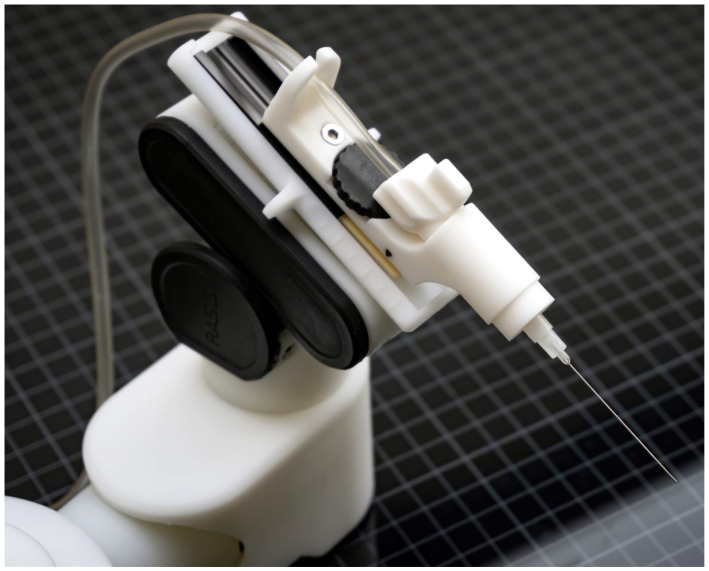
The employed custom-made robot holding an injection needle.

**Figure 2 micromachines-14-01261-f002:**
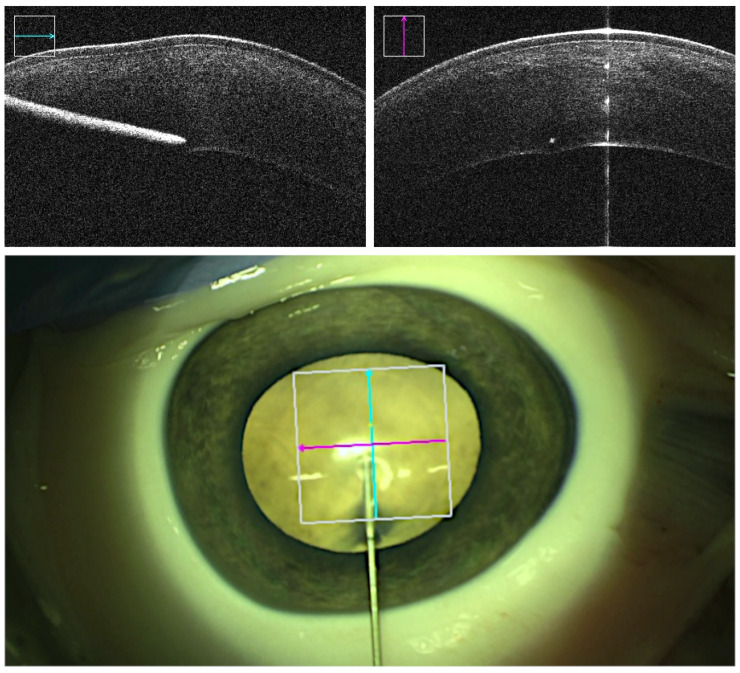
OCT acquisitions (**top**) and the microscopic view (**bottom**) from a robot-assisted trial immediately before air injection. The cyan and magenta arrows indicate OCT capture locations. The cannula tip is placed deep into the stroma while avoiding perforation into the DM.

**Figure 3 micromachines-14-01261-f003:**
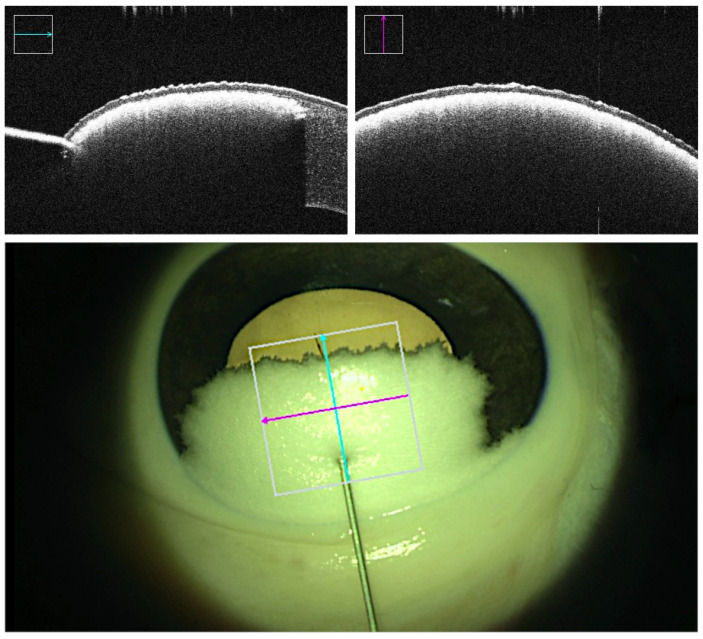
An example of a failure in deep air injection. OCT acquisitions (**top**) and the microscopic view (**bottom**) show superficial intrastromal bubble formation that appears as a cloudy layer directly below the corneal epithelium. The cyan and magenta arrows indicate OCT capture locations.

**Figure 4 micromachines-14-01261-f004:**
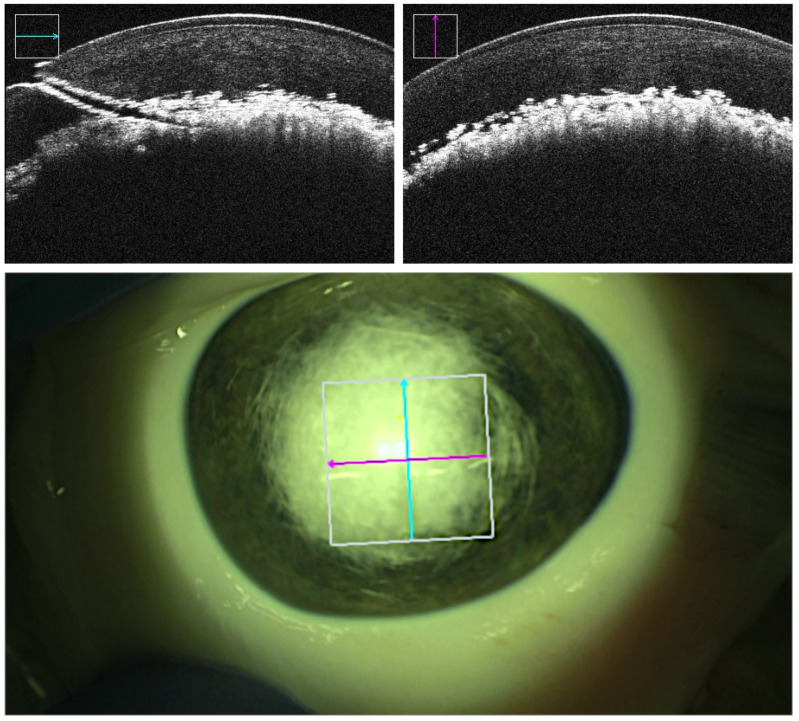
An example of a successful deep air injection visible in OCT acquisitions (**top**) and the microscopic view (**bottom**). The tunnel formed by the cannula is apparent in the OCT B-scan captured along the cannula trajectory (**top left**). The cyan and magenta arrows indicate OCT capture locations.

**Figure 5 micromachines-14-01261-f005:**
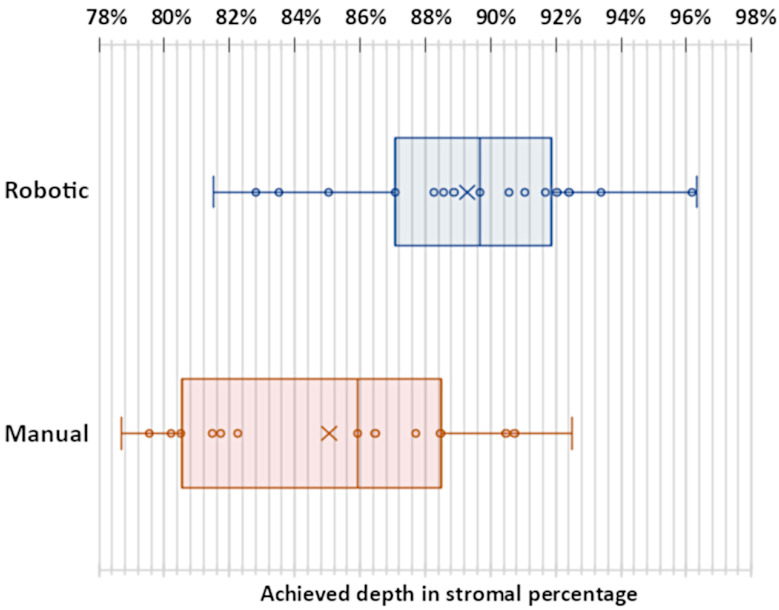
The box plot showing the reached depths in successful manual and robot-assisted trials. The inner band and cross represent the median and the mean, respectively.

**Table 1 micromachines-14-01261-t001:** Means and standard deviations of the measured parameters in different settings.

		Successful Manual (74%)	Failed Manual (26%)	Successful Robotic (83%)	Failed Robotic (17%)
Achieved depth (stromal percentage)	Mean STD	85.05% 4.52%		89.29% 4.03%	
Needle entry angle (degree)	Mean STD	23.45° 8.80°	28.61° 9.13°	23.35° 6.48°	32.44° 5.73°
Needle tunnel length (millimeters)	Mean STD	1.82 mm 0.27 mm		1.72 mm 0.45 mm	
Operation duration (seconds)	Mean STD	212 s 96 s		260 s 96 s	

## Data Availability

No applicable.

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
