# Peer review of "Comparison of Robot-Assisted and Manual Cannula Insertion in Simulated Big-Bubble Deep Anterior Lamellar Keratoplasty"

_micromachines, 2023, doi:10.3390/mi14061261_

Round 1

Reviewer 1 Report

In this manuscript, the authors describe the use of intraoperative OCT and a piezo based robot to insert cannulas into porcine eyes to simulate big bubble DALK procedures. Once inserted, pneumodissection was used in an attempt to generate a big bubble in the porcine eye. The resulting needle depth and stromal demarcation plane were compared to the results from the same surgeons performing the DALK technique using a manual approach. This reviewer acknowledges the technical difficulty of the DALK procedure and agrees that robotic assistance with OCT guidance is one potential solution to overcome the surgical challenges of the technique.

Overall, this manuscript is laid out well and easy to follow. The successful use of robotic assistance for needle and cannula guidance in other fields indicates this strategy might be useful for DALK procedures. Below are comments and suggestions that should be addressed to clarify and strengthen the manuscript.

Introduction:

·         The authors do well to motivate the technical difficultly of the DALK procedure and highlight two key factors by which the DALK procedure can be improved (1. OCT for depth guidance, and robot-assistance to 2. remove hand tremor). The reviewer appreciates the breadth of the prior works listed, however it is confusing that the same references are attributed to different authors in the text. Specifically references [20], and [21].  Please be consistent in your citation.

·         There should be a space between “over” and “80%” on line 57%.

Materials and Methods:

2.2 Robot

·         It is unclear how the robot is mounted in the experiments (Is it on the surgical table, or is it intended to be mounted on the patient). It is also unclear what the mechanical limits of the robot are.  This section could benefit from more details on the position of the robot with respect to the microscope and patient, as well as description of the surgical volume that can be reached by the robot as well as the mechanical limits of each joint.

·         The user controllable features of the robot are unclear. For instance, is a clutch feature needed/used to account for mechanical motions that are beyond the range of motion of the 3D mouse?  Does the robot have motion scaling capabilities, i.e. does 1cm of joystick motion correlate to 1cm of robot motion and can this be scaled?  Is there a perceived delay between motion of the joystick and the response of the robot? More details would be beneficial to the reader.

·         This section should include a reference to Section 2.5 Method indicating to the reader how the initial position of the robot is selected to guarantee the needle is correctly aligned in the surgical workspace.

·         Figure 1 would benefit from identifying the DOF of the robot with arrows showing the respective motions of each joint.

2.5 Method

·         Please clarify how many trials and how much time each novice surgeon was allowed to have before beginning the main study.

·         There authors state a total of 48 trials performed by 2 novice surgeons indicating a total of 24 procedures completed by each. How did the study account for the learning curve of each surgeon as they performed more trials? It would seem that the performance during the final samples for each surgeon would be markedly better than the first samples, especially since the surgeons had no prior experience with DALK before the study. 

·         Please also clarify if surgeons performed all manual samples first followed by all robotic, or if the order of samples was random since the order in which samples are performed could have an impact on the results with the use of novice surgeons.  I.e. Did surgeons perform better on later samples and worse on the first samples and could this be a factor in the differences between the robotic and manual study arms.?

·         Please clarify and justify the typed of t-test used (paired versus unpaired).

Additional comments:

·         It is unclear how the Porcine eyes are prepared for this experiment, the health of the porcine tissue, and time postmortem.  The authors also state that some of the eyes did not have enough intraocular pressure, but it is unclear how pressure was measured or set to a value for the experiments.  These factors impact the ability of a user (or robot) to insert needles, as well as the thickness of the cornea during experiments. Please add details on how the eyes were obtained, how old the eyes were, how the eyes were prepared for experiments so the reader can better understand the quality of the porcine eyes and the experiments could be repeated.

Discussion:

·         Please expand on why you believe robotic needle insertion was significantly longer than the manual needle insertion, as this seems counter intuitive especially for novice surgeons performing the DALK procedure for the first time.

·         Pneumodissection and resulting demarcation of the tissue can be impacted by the relative angle of cannula insertion. It appears that there may be a significant difference between the angle of insertion for successful trials versus unsuccessful trials.  Please discuss what factors caused such differences in the insertion angle, if these differences are significant, and how these differences could impact differences in demarcation layers between successful and unsuccessful trials. 

Well written paper with only minor language and grammar concerns.

Author Response

Dear Editors and Referees,

Thank you very much for your constructive and helpful reviews on our manuscript entitled “Comparison of robot-assisted and manual cannula insertion in simulated big-bubble deep anterior lamellar keratoplasty “. The following is a summary of some modification details.

  1. “Materials and Methods”
  • Detailed descriptions of robotic aids have been added.
  • The method of obtaining the relevant parameters of the experimental material and the reason for the establishment of the experimental model have been added.
  1. “Discussion”
  • Explained and analysed the results obtained in the experiment that "the duration of the robot-assisted experimental operation is slightly longer than that of the manual experimental operation".
  • Added description of potential risks of robotic assistance systems.
  1. “Introduction and Results”
  • Added detailed explanations for some technical terms.

Thanks a lot for your consideration.

PD Dr. -Ing. Habil. med. M. Ali Nasseri

Ophthalmology Department of Klinikum rechts der Isar, Technische Universität München, Munich, Germany, 81675.

E-mail: ali.nasseri@mri.tum.de

Reviewer 2 Report

Although I am not a specialist in ophtalmologic surgeries, I believe that the current study can be a pilot article in the respective field.

I would have one comment: please add a paragraph on what would the complications/ risks of adopting robotic assistance in cannula placement?

Thank you.

Good.

Author Response

(The authors gave the same response as above.)
